# REFORM : Residual Filtering through Neural Aggregators for Layer-Wise Representation Integrity

## Abstract

Recent studies suggest that cumulative residual connections in Transformer-based LLMs preserve signals indiscriminately, potentially leading to representation bottlenecks in deeper layers. In this work, we provide an information-theoretic analysis of this phenomenon and introduce REFORM (Representation formulation via multi-layer aggregation), a lightweight, training-only module designed to address this issue. REFORM hierarchically integrates hidden states across layers, leveraging local aggregation for continuity and global fusion for semantic abstraction. During training, three auxiliary objectives—correlation alignment, orthogonality, and cosine similarity—guide REFORM to restructure intermediate representations. Notably, REFORM is detached at inference time, thus incurring no runtime overhead. Extensive evaluations across Llama3, Qwen2, Mistral, and Phi-3.5 models on commonsense and mathematical reasoning benchmarks demonstrate consistent improvements. Analysis using SVCCA, attention entropy, and effective rank suggests that REFORM fosters richer representations, especially in mid-to-late layers, indicating that minimal inter-layer aggregation can help alleviate structural limitations without sacrificing inference efficiency. We provide the code at https://anonymous.4open.science/r/Reform.

## 1 Introduction

Large language models (LLMs) based on Transformer (Vaswani et al., 2017) architecture have become a dominant paradigm in natural language processing (NLP) (Bai et al., 2023; Liu et al., 2024; Touvron et al., 2023a;b; Grattafiori et al., 2024; Achiam et al., 2023). These models follow a hierarchical structure where hidden states are sequentially propagated across layers. To stabilize optimization and preserve information flow, residual connection (RC) (He et al., 2016) was introduced. However, recent studies (Zhu et al., 2025; Liu et al., 2020; Barbero et al., 2024a; Li & Papyan, 2023; Arefin et al., 2024; Gerasimov et al., 2025) show that RC may not effectively propagate information. Since RC adds input and output directly, it can overlook semantic changes across layers and reduce representational diversity, contributing to a *representation bottleneck*[1]. Probing studies (Fan et al., 2024; Ben-Artzy & Schwartz, 2024; Jawahar et al., 2019; Pirozelli et al., 2024; Van Aken et al., 2019) further reveal that Transformer layers encode distinct features, suggesting that simple summation risks obscuring task-relevant information and suppressing diversity.

From this perspective, we argue that effective information processing and the maintenance of representational diversity are key to improving LLM performance. To characterize attention dispersion, we report mean attention entropy (MAE) (Ghader & Monz, 2017), which measures the uniformity of attention distributions across heads. Concretely, MAE is the average entropy of attention weights: lower values indicate more focused attention on specific tokens, whereas higher values reflect broader, more uniform distributions across token positions. We treat MAE as an auxiliary attention-dispersion proxy and do not equate it with representational diversity.

Figure 1 presents the layer-wise MAE trends across Llama1 (Touvron et al., 2023a) to Llama3 (Grattafiori et al., 2024). From Llama1 to Llama2, overall MAE increases, indicating

---

[1]We use *representation bottleneck* to emphasize structural limitations of residual connections, rather than strict collapse.

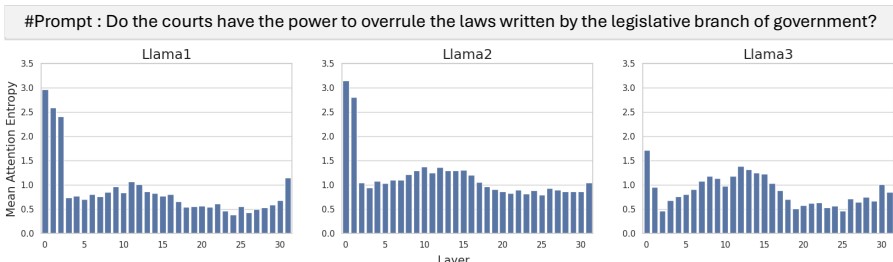

Figure 1: Mean attention entropy (MAE) values across all layers for Llama1, Llama2, and Llama3, based on a single prompt: "Do the courts have the power to overrule the laws written by the legislative branch of government?". Further examples are provided in Appendix A.

*more distributed attention*; while MAE reflects dispersion (not diversity), here it aligns with our co-variance/SVCCA trends. Llama3 shows slightly lower overall MAE than Llama2 but a more dynamic profile with notably high mid-layer entropy. This suggests that representational capacity depends not only on global entropy level but also on layer-wise variation; a purely additive residual mechanism that treats layers uniformly may miss these layer-specific contributions. Consequently, what appears as a "representation bottleneck" is not just reduced diversity but a weakening of information flow along the residual path, where critical features are gradually diluted. Viewed through the information bottleneck lens (Tishby et al., 2000), this motivates revisiting residual design toward mechanisms that preserve task-relevant structure while enabling selective compression.

To this end, we propose REpresentation FORmulation via Multi-layer aggregation (REFORM), which hierarchically processes and integrates hidden states from all Transformer layers. REFORM employs two hierarchical aggregators that separately model local continuity across adjacent layers and global abstraction across the full stack: local continuity supports fine-grained selection, while global abstraction supports cross-layer integration. By explicitly capturing continuity and inter-layer interactions, REFORM mitigates distortions in information flow and preserves layer-specific characteristics. The module is used only during training and removed at inference, adding no runtime overhead. Experiments show consistent gains, especially on complex tasks, underscoring the value of aggregation strategies that account for both inter-layer information flow and layer-specific structure. Our key contributions are summarized as follows:

- We present a systematic analysis of residual connection as a cumulative addition process and rein-terpret their effect on deep representations through the lens of the information bottleneck, showing how existing structure with RC can hinder selective compression and semantic disentanglement.

- We propose REFORM, a novel module that explicitly preserves and utilizes layer-wise semantic features via hierarchical aggregation.

- We demonstrate that REFORM yields consistent improvements across multiple LLM architectures and tasks, supporting its generality and robustness.

## 2 RELATED WORK

Following the success of ResNet (He et al., 2016), Residual connection (RC) has been widely adopted to alleviate gradient vanishing and improve training stability in deep neural networks, particularly in Transformer-based models and large language models (LLMs) (Bai et al., 2023; Liu et al., 2024; Achiam et al., 2023; Mann et al., 2020; Team et al., 2023; Chung et al., 2024; Raffel et al., 2020). While RC eases information flow, recent studies (Zhu et al., 2025; Liu et al., 2020; Li & Papyan, 2023; Arefin et al., 2024; Gerasimov et al., 2025; Barbero et al., 2024b; Wang et al., 2024b) show it can miss hierarchical structure, reducing diversity and causing a *representation bottleneck*. To address these issues, various techniques (Zhu et al., 2025; Liu et al., 2019; Wang et al., 2024a) have been proposed, aiming to improve RC by adjusting weights or stabilizing gradients. However, these extensions do not explicitly explore the structural limitations inherent in simple addition mechanism of RC. (Li & Papyan, 2023) identify such limitations in the context of CNNs, and our work extends this discussion to Transformer-based LLMs by reinterpreting them from a *representation bottleneck* perspective. Specifically, we highlight that simple addition in residual paths may dilute hierarchical information, and we propose a training-time module designed to recover this information.

In particular, effectively integrating hierarchical representations across layers is crucial for enhancing the model's expressiveness. Prior studies (Ju et al., 2024; Jiang et al., 2024) have shown that knowledge tends to be compressed in shallow layers and progressively expanded to other tokens in deeper layers. Furthermore, (Jawahar et al., 2019; Jin et al., 2025; Sajjad et al., 2022) have shown that shallow layers primarily capture entity-level features, whereas deeper layers encode higher-level semantic information. Recent findings (Fan et al., 2024; Ben-Artzy & Schwartz, 2024; Jawahar et al., 2019; Pirozelli et al., 2024; Van Aken et al., 2019) also indicate that each layer captures different features of task-specific representations. These findings emphasize the significance of leveraging hierarchical representations across layers. Motivated by this, we propose a novel framework that aggregates layer-wise features and captures semantic transitions across layers.

## 3 REPRESENTATION BOTTLENECK: AN INFORMATION-THEORETIC REVISIT

**Observations.** Figure 1 shows that Llama3's mean attention entropy (MAE) drops sharply at layer 3, rises mildly over layers 4–10, and declines again across layers 11–25, suggesting reduced diversity of attention patterns. Inter-layer similarity via SVCCA (Raghu et al., 2017) concurrently increases in mid depths (Fig. 2; layers 20–25 > 0.901), indicating redundancy among layer representations. In the final layers (30–32), MAE rises while SVCCA decreases, hinting at an architectural compensation that restores some expressivity near decoding.

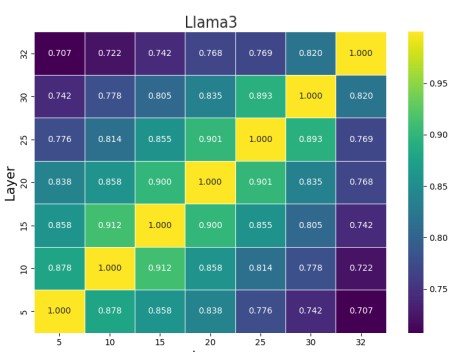

Figure 2: Layer-wise SVCCA in Llama3

*Why does representation expressivity saturate in mid-to-deep layers?*

**Operating regime and scope.** We analyze the residual update on a mid-to-deep depth interval $L \subset \{1, \ldots, L_{\max}\}$ characteristic of late training (strong skip dominance):

$$H^{(l)} = H^{(l-1)} + F^{(l)}(H^{(l-1)}), \qquad H^{(l)} \in \mathbb{R}^D. \tag{1}$$

The analysis uses second-order statistics; full Gaussianity is not required. We use covariance matrices $\Sigma \succeq 0$ (positive semidefinite, PSD; $\succeq$ denotes the Loewner order). Empirical studies report that, late in training, residual updates align with their inputs (Li & Papyan, 2023); the cosine similarity between $F^{(l)}(H^{(l-1)})$ and $H^{(l-1)}$ increases with depth/time, drifting toward near-identity updates. This observed residual alignment motivates the near-identity decomposition $A_l = \rho_l I + E_l$ and predicts diminishing per-layer spectral changes in late training.

**Support stability (purpose and assumption).** Along $L$, we assume the support does not change: $\text{rank}(\Sigma_{l-1}) = r$ and the smallest positive eigenvalue satisfies $\lambda^+_{\min}(\Sigma_{l-1}) \geq \tau > 0$. This avoids rank flips within $L$ and ensures stable pseudo-determinant expansions; it prevents spurious rank changes that would make $\log \det^+$ ill-defined across $L$.

**Assumptions (near-identity residual regime).** For $l \in L$:

(A1) **Local linearization with small gain.** There exists $A_l \in \mathbb{R}^{D \times D}$ such that

$$F^{(l)}(H^{(l-1)}) = A_l H^{(l-1)} + \varepsilon_l(H^{(l-1)}),$$

with layer gain $\epsilon_l := \|A_l\|_2 \ll 1$ and a remainder quadratically small on average:

$$\mathbb{E}\|\varepsilon_l(H^{(l-1)})\|_2 \leq c_\varepsilon \, \epsilon_l^2 \, \mathbb{E}\|H^{(l-1)}\|_2,$$

for a constant $c_\varepsilon > 0$ uniform in $l \in L$.

(A2) **Near-identity alignment with bounded anisotropy.** Decompose $A_l = \rho_l I + E_l$, with

$$\rho_l := \frac{\mathbb{E}\langle H^{(l-1)}, F^{(l)}(H^{(l-1)})\rangle}{\mathbb{E}\|H^{(l-1)}\|_2^2}, \qquad \eta_l := \|E_l\|_2 \ll 1.$$

Assume: (i) *First-order support preservation*: $\text{Tr}(P_{l-1}E_l) = 0$, where $P_{l-1} := \Sigma_{l-1}\Sigma_{l-1}^+$; (ii) *Admissible gain*: $\rho_l \in [-1+\delta, \rho_{\max}]$ for fixed $\delta > 0$.

(A3) **Input regularity.** $H^{(l-1)}$ has zero mean, covariance $\Sigma_{l-1} \succeq 0$, and subgaussian (or elliptical) tails; second moments exist and are finite.

**Proxies.** To quantify spectral evolution under (A1)–(A3), we use proxies that depend only on the positive spectrum of $\Sigma$ and are stable under fixed rank:

(i) **Entropy and effective rank:** for normalized spectrum $p_i = \lambda_i / \sum_{\lambda_j > 0} \lambda_j$, define $\mathcal{H}(p) := -\sum_i p_i \log p_i$ and $r_{\text{eff}}(\Sigma) := \exp(\mathcal{H}(p))$.

(ii) **Log pseudo-determinant:** $\log \det^+(\Sigma) = \sum_{\lambda_i > 0} \log \lambda_i = \text{Tr}(\log(\Sigma|_{\text{supp}(\Sigma)}))$, where $\text{supp}(\Sigma) := \text{range}(\Sigma)$ and $\Sigma|_{\text{supp}(\Sigma)} := P\Sigma P$ with $P = \Sigma\Sigma^+$ the projector onto $\text{supp}(\Sigma)$.

**Core structural relation.** From local linearization and the decomposition $A_l = \rho_l I + E_l$, together with the nonlinear remainder, there exists a remainder $\Delta_l$ (arising from the anisotropic part $E_l$ and higher-order terms) such that

$$\Sigma_l = (1+\rho_l)^2 \Sigma_{l-1} + \Delta_l, \qquad \|\Delta_l\|_2 \le C\left(\eta_l + \rho_l \eta_l + \epsilon_l^2\right)\|\Sigma_{l-1}\|_2,$$

where $C$ is uniform in $l \in L$. Define the *support–relative perturbation*

$$M_l := (\Sigma_{l-1}^+)^{1/2}\, \Delta_l\, (\Sigma_{l-1}^+)^{1/2}.$$

Intuitively, $M_l$ measures the perturbation *relative to the support of* $\Sigma_{l-1}$: isotropic scaling and null-space effects are removed, and spectral quantities such as $\log \det^+$ or $r_{\text{eff}}$ vary at most $O(\|M_l\|_2)$.

**Main lemma.** Under (A1)–(A3), support stability ($\lambda_{\min}^+ \ge \tau$), and sufficiently small ($\eta_l, \epsilon_l$), the per-layer changes satisfy

$$r_{\text{eff}}(\Sigma_l) = r_{\text{eff}}(\Sigma_{l-1}) + O(\|M_l\|_2), \quad \log \overset{+}{\det}(\Sigma_l) = \log \overset{+}{\det}(\Sigma_{l-1}) + 2r\log(1+\rho_l) + O(r\|M_l\|_2),$$

where $\|M_l\|_2 \le C(\eta_l + \rho_l \eta_l + \epsilon_l^2)$. Combining yields

$$r_{\text{eff}}(\Sigma_l) = r_{\text{eff}}(\Sigma_{l-1}) + O\left(\eta_l + \rho_l\eta_l + \epsilon_l^2\right),$$

$$\log \overset{+}{\det}(\Sigma_l) = \log \overset{+}{\det}(\Sigma_{l-1}) + 2r\log(1+\rho_l) + O\left(r\left(\eta_l + \rho_l\eta_l + \epsilon_l^2\right)\right).$$

Here the $\eta_l^2$ term from $E_l \Sigma E_l^\top$ is absorbed into $\eta_l$ since $\eta_l \ll 1$.

**Implications.** Isotropic amplification $(1 + \rho_l)^2$ leaves the spectrum's *shape* (hence $r_{\text{eff}}$) unchanged, while anisotropy $E_l$ contributes at second order; thus entropy/effective-rank gains are marginal per layer and plateau in mid depths, matching SVCCA rise and MAE decline. Details in Appendix B.

## 4 METHOD

In Transformer-based large language models (LLMs), each layer captures layer-specific characteristics, but the simple cumulative addition through residual connections does not fully reflect the hierarchical representations. In light of this limitation, we introduce REFORM, a framework that extracts hidden states from all Transformer layers and processes them in a hierarchical manner. Figure 3 illustrates the overall architecture of the proposed model. REFORM augments a vanilla Transformer with two hierarchical modules: Local Aggregation across adjacent layers and Global Aggregation across all layers. This design captures both local continuity and global semantic.

First, given an input sequence $X \in \mathbb{R}^{T \times D}$, where $T$ is the number of tokens in the input sequence and $D$ is the hidden dimension of each token embedding, we extract hidden states $H^l$ from each of the $L$ layers. For a given token position $t$, the hidden states at each layer are defined as follows:

$$H_t = [H_t^1, H_t^2, ..., H_t^L] \in \mathbb{R}^{L \times D}, \tag{2}$$

where $H_t^l$ denotes the hidden state at layer $l$. Dividing the representation into $K$ attention heads of size $d_K = D/K$, we reshape:

$$H_t \in \mathbb{R}^{L \times K \times d_K}. \tag{3}$$

These stacked hidden states serve as input to REFORM, enabling hierarchical modeling across layers.

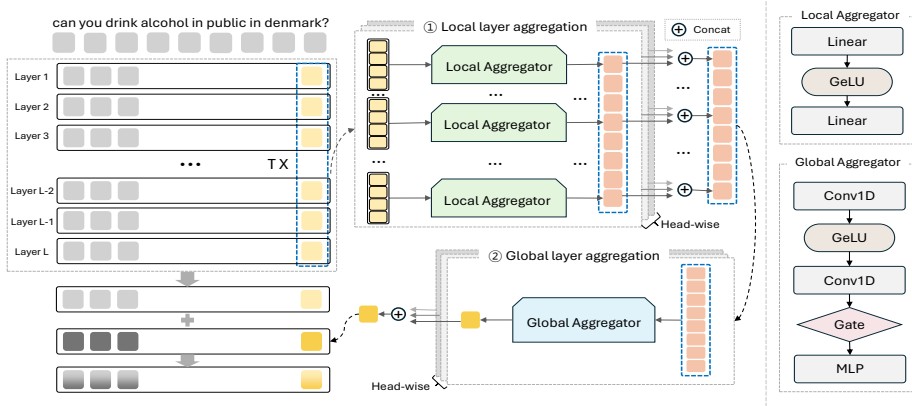

Figure 3: Overview of the proposed REFORM framework.

## 4.1 AGGREGATOR

The proposed Aggregator receives grouped hidden states as input and models layer-wise variations per attention head. It operates independently in both local and global aggregation, thereby capturing hierarchical representations within the Transformer.

The grouped hidden state is split into $K$ attention heads along the feature dimension, and each head is processed by an independent module. This design follows the principle that multiple attention heads capture distinct subspaces, enabling each to learn transformations without parameter sharing. Such head-wise modularity encourages disentanglement of semantic components across layers.

Formally, for the hidden state $h_t^{(g)}[k]$ of the $k$-th head at position $t$ in group $g$, the aggregation is:

$$z_t^{(g)}[k] = \text{Aggregator}_k\left(h_t^{(g)}[k]\right), \quad k = 1, \ldots, K, \tag{4}$$

and the group-level representation is then obtained by concatenation:

$$z_t^{(g)} = \text{Concat}(z_t^{(g)}[1], ..., z_t^{(g)}[K]). \tag{5}$$

Each Aggregator block adopts either a linear or convolution-based module, with shallow structures maintained for parameter efficiency and training stability.

## 4.2 LOCAL LAYER AGGREGATION

Adjacent Layers often exhibit smooth, semantically correlated changes. To capture these local transitions, we divide the $L = 32$ layers into eight non-overlapping groups of four consecutive layers:

$$(H_t^{(1)}, ..., H_t^{(8)}) = (H_t^{1:4}, H_t^{5:8}, ..., H_t^{29:32}). \tag{6}$$

Each group $H_t^{(g)}$ is then passed through a Local Aggregator, implemented as a linear operation:

$$Z_t^{(g)} = \text{Local Aggregator}_k(H_t^{(g)}), \quad k = 1, ..., K, \tag{7}$$

This produces group-level representations $Z_t^{(g)}$ encoding short-range transitions. Local aggregation acts as a fine-grained selector, highlighting inter-layer differences within each group.

## 4.3 GLOBAL LAYER AGGREGATION

To capture abstract representations across the full layer stack, the eight local representations $Z_t = \{Z_t^{(1)}, ..., Z_t^{(8)}\}$ are further processed by a Global Aggregator. Implemented as a head-wise 1D convolution, this stage integrates layer-group information into the final token-level representation:

$$R_t = \text{Global Aggregator}_k(Z_t), \quad k = 1, ..., K. \tag{8}$$

Global aggregation provides high-level semantic abstraction over the hierarchy. Combined with local selection, the two-stage design promotes semantic staging—select first, abstract later—addressing the limitations of cumulative residual connection.

## 4.4 TRAINING AND INFERENCE

Finally, the global representation $R_t$ is integrated with the final-layer hidden state $H_t^L$ of the vanilla Transformer for prediction. We conceptualize them as complementary channels: $R_t$ encodes staged hierarchical abstractions, while $H_t^L$ preserves the original signal trajectory. Their integration through a learnable gate enables adaptive fusion of structural and residual representations. To encourage the two representations to learn complementary information, we define the following training objectives:

### 4.4.1 ORIGINAL LOSS

$\mathcal{L}_{base}$ denotes the standard cross-entropy loss used in LLMs, applied to predict the target token from the final representation. Specifically, $R_t$ and $H_t^L$ are integrated using a gating mechanism to produce the final representation $\tilde{H}_t$, which is then fed into the loss function:

$$\tilde{H}_t = \gamma_t \odot R_t + H_t^L, \tag{9}$$

$$\gamma_t = \sigma(W H_t^L + b), \tag{10}$$

where $\gamma_t$ is a learnable gate vector that modulates the output of the aggregator, serving as a compensation to the vanilla representation $H_t^L$.

### 4.4.2 AUXILIARY LOSSES

Training solely with $\mathcal{L}_{base}$ risks $R_t$ acting as noise or collapsing into a meaningless representation. To prevent these and encourage $R_t$ to encode meaningful information that complements $H_t^L$, we introduce the following three auxiliary losses during training. These auxiliary losses are applied only during training and removed during inference, where predictions rely solely on the vanilla Transformer without the aggregator.

Furthermore, we adopt a stochastic activation strategy (Medvegyev, 2007), where REFORM is probabilistically activated during training. This strategy improves robustness by ensuring stable performance even when the aggregator is removed at inference. By design, REFORM acts as a training-time-only auxiliary, allowing the model to explore richer intermediate representations while maintaining inference-time efficiency.

**Correlation Alignment Loss $\mathcal{L}_{ca}$.**  To prevent the two representations from diverging while ensuring consistent predictive direction, we adopt the correlation alignment loss (Chandar et al., 2016) $\mathcal{L}ca$. This loss maximizes the cosine similarity between $R_t$ and $H_t^L$ after removing the mean from each representation. $R_t$ learns to preserve variation while remaining aligned with $H_t^L$. During this process, the gradient flow through $H_t^L$ is blocked, ensuring $R_t$ follows the directional guidance of $H_t^L$ without directly copying it. This encourages $R_t$ to develop an independent representation.

$$\mathcal{L}_{ca} = 1 - \frac{1}{T} \sum_{t=1}^{T} \cos(R_t - \text{mean}(R_t), H_t^L - \text{mean}(H_t^L)). \tag{11}$$

**Orthogonal Loss $\mathcal{L}_{orth}$.**  To reduce redundancy and encourage complementary features, we adopt an orthogonal regularization term from (Wortsman et al., 2021). Specifically, we square the cosine similarity between $R_t$ and $H_t^L$, which (1) treats positive and negative correlations equally and (2) grows quadratically with $|\cos \theta|$, thus penalizing stronger alignments more heavily. The gradient through $H_t^L$ is blocked so that only $R_t$ adapts to reduce redundancy:

$$\mathcal{L}_{orth} = \frac{1}{T} \sum_{t=1}^{T} \cos(R_t, H_t^L)^2. \tag{12}$$

**Cosine Similarity Loss $\mathcal{L}_{cs}$.**  $\mathcal{L}_{cs}$ facilitates removal of the aggregator at inference. In this objective, $\tilde{H}_t$ is used with its gradient flow blocked, and $H_t^L$ is trained to approximate $\tilde{H}_t$. This alignment encourages $H_t^L$ to support stable predictions even without the aggregator. Unlike $\mathcal{L}_{ca}$, which enforces directional consistency via mean-subtracted alignment, $\mathcal{L}cs$ enforces full-vector alignment, enabling direct substitution at inference.

$$\mathcal{L}_{cs} = 1 - \frac{1}{T} \sum_{t=1}^{T} \cos(\tilde{H}_t, H_t^L). \tag{13}$$

Together, these objectives support representation disentanglement ($\mathcal{L}_{orth}$), alignment ($\mathcal{L}_{ca}$), and inference-time consistency ($\mathcal{L}_{cs}$). The overall training objective is defined as:

$$\mathcal{L} = \mathcal{L}_{base} + \lambda_{orth} \cdot \mathcal{L}_{orth} + \lambda_{ca} \cdot \mathcal{L}_{ca} + \lambda_{cs} \cdot \mathcal{L}_{cs}. \tag{14}$$

Here, $\lambda_{ca}$ is fixed at 0.1, while $\lambda_{orth}$ and $\lambda_{cs}$ follow a cosine bump schedule—a symmetric strategy that keeps weights low at the beginning and end of training, increasing them in the middle. This enables auxiliary losses to contribute effectively, maintaining overall training stability.

## 5 EXPERIMENTS

### 5.1 EXPERIMENTAL SETTING

To validate the generality and effectiveness of the proposed REFORM, we consider four open-source LLMs from different model families: `Llama3-8B-Instruct`, `Qwen2-7B-Instruct`, `Mistral-7B-Instruct-v0.3`, and `Phi-3.5-Mini-Instruct`. For the experiments, we use two Nvidia A100 80GB GPUs. Results are averaged over 5 random seeds across all benchmarks.

**Datasets.** We conduct extensive evaluations of REFORM on multiple benchmarks and compare the results with existing baselines. The evaluation is organized into two major categories: Commonsense Reasoning and Arithmetic Reasoning. A detailed description and hyperparameter settings of these benchmarks are provided in Appendix C.

Table 1: Commonsense reasoning

| Method | OBQA | ARC-e | ARC-c | WinoG. | PIQA | BoolQ | HellaS. | Avg. |
|---|---|---|---|---|---|---|---|---|
| Llama3 + LoRA | 84.9 | 90.2 | 79.9 | **85.5** | 87.9 | 69.9 | 94.4 | 84.7 |
| + Ours | **85.3 (+0.4)** | **90.3 (+0.1)** | **80.5 (+0.6)** | 85.1 (-0.4) | **88.7 (+0.9)** | **71.8 (+1.9)** | **94.9 (+0.5)** | **85.2 (+0.5)** |
| Qwen2 + LoRA | **89.3** | 92.7 | 83.1 | 85.5 | 89.5 | 73.9 | 94.6 | 86.9 |
| + Ours | 89.1 (-0.2) | **93.2 (+0.5)** | **84.2 (+1.1)** | **85.7 (+0.2)** | **89.8 (+0.3)** | **74.1 (+0.2)** | **94.8 (+0.2)** | **87.3 (+0.4)** |
| Mistral + LoRA | 84.0 | 85.7 | 73.4 | 83.0 | 87.4 | 71.0 | 90.3 | 82.1 |
| + Ours | **85.4 (+1.4)** | **86.1 (+0.4)** | **74.6 (+1.2)** | **84.5 (+1.5)** | **87.5 (+0.1)** | **71.8 (+0.8)** | **90.7 (+0.4)** | **82.9 (+0.8)** |
| Phi-3.5 + LoRA | 88.6 | 94.4 | 85.4 | 83.0 | 86.1 | 70.3 | 90.8 | 85.5 |
| + Ours | **89.3 (+0.7)** | **94.7 (+0.3)** | **86.0 (+0.6)** | **83.6 (+0.6)** | **86.8 (+0.7)** | **70.5 (+0.2)** | 90.8 (±0.0) | **86.0 (+0.5)** |

Table 2: Arithmetic reasoning

| Method | SVAMP | AddSub | AQuA | MultiArith | GSM8K | Avg. |
|---|---|---|---|---|---|---|
| Llama3 + LoRA | 82.7 | **92.9** | 32.3 | 98.0 | 72.9 | 75.8 |
| + Ours | **83.8 (+1.1)** | 92.9 (±0.0) | **34.3 (+2.0)** | **98.3 (+0.3)** | **73.6 (+0.7)** | **76.6 (+0.8)** |
| Qwen2 + LoRA | 82.8 | **91.6** | 36.9 | **97.3** | 73.9 | 76.5 |
| + Ours | **83.5 (+0.7)** | 91.2 (-0.4) | **37.6 (+0.7)** | 97.3 (±0.0) | **75.0 (+1.1)** | **76.9 (+0.4)** |
| Mistral + LoRA | 65.3 | 87.9 | 23.7 | 95.0 | 62.6 | 66.9 |
| + Ours | **66.8 (+1.5)** | **88.9 (+1.0)** | **24.7 (+1.0)** | **95.3 (+0.3)** | **63.2 (+0.6)** | **67.8 (+0.9)** |
| Phi-3.5 + LoRA | 76.4 | **86.1** | 33.6 | 98.1 | 76.0 | 74.0 |
| + Ours | **76.9 (+0.5)** | 85.5 (-0.6) | **34.1 (+0.5)** | **98.4 (+0.3)** | **77.0 (+1.0)** | **74.4 (+0.4)** |

### 5.2 MAIN RESULTS

Tables 1 and 2 summarize REFORM's performance on commonsense and arithmetic reasoning benchmarks. REFORM shows consistent improvements on most benchmarks, particularly on structurally complex tasks such as ARC-c, HellaSwag, AQuA, and SVAMP. For example, ARC-c improves by up to 1.2% (Mistral), PIQA by 0.9% (Llama3), and AQuA by 2.0% (Llama3). Gains on simpler tasks such as ARC-e are also observed, though they remain modest.

Some tasks show minimal gains or slight regressions. Qwen2 drops by 0.2% on OBQA, and Llama3 underperforms on WinoGrande (–0.4%), possibly due to perturbations in calibrated decision boundaries of already confident models. A similar effect is observed in Qwen2 (–0.4%) and Phi-3.5 (–0.6%) on AddSub, where representational shifts may disrupt low-entropy outputs.

Overall, these results suggest that REFORM is most effective on benchmarks requiring multi-step reasoning or narrative consistency, such as ARC-c, HellaSwag, AQuA, and SVAMP. In contrast, tasks

Table 3: Component comparison on Local and Global Aggregator architectures.

| Local | Global | Avg. |
|---|---|---|
| Linear | Linear | 76.3 |
| Linear | 1D-Conv | **76.6** |
| 1D-Conv | Linear | 76.1 |
| 1D-Conv | 1D-Conv | 75.9 |

Table 4: Ablation study on auxiliary losses for the Arithmetic benchmark.

| Objective | Avg. |
|---|---|
| $\mathcal{L}_{total}$ | **76.6** |
| w/o $\mathcal{L}_{orth}$ | 76.0 |
| w/o $\mathcal{L}_{ca}$ | 76.4 |
| w/o $\mathcal{L}_{cs}$ | 75.8 |

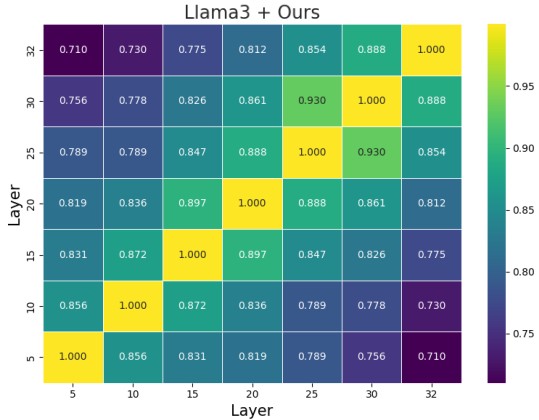

Figure 4: Layer-wise SVCCA in Llama3 + Ours.

with shallow decisions or simple arithmetic (e.g., ARC-e, AddSub) show smaller gains, indicating that REFORM's hierarchical integration provides the greatest benefit when structural abstraction is required. Detailed results with mean and standard deviation are reported in Appendix D.1.

## 5.3 ABLATION STUDY AND COMPONENT COMPARISON

Table 3 shows the performance of different combinations of Local and Global Aggregators. The highest score observed (76.6%) is achieved with a Linear–Conv setup, where local selection precedes global integration. This may be attributed to semantic staging: the Linear module filters salient directions, while the 1D Conv captures contextual coherence across layers.

Table 4 reports ablations on auxiliary losses. Removing each loss individually reduces performance, with $\mathcal{L}_{cs}$ causing the largest drop ($-0.8\%$), underscoring its importance for stable inference. $\mathcal{L}_{orth}$ and $\mathcal{L}_{ca}$ also contribute to representational diversity and alignment. Together, the three losses provide complementary benefits for robustness and generalization. Additional ablations and qualitative analyses are presented in Appendix D.2 and Appendix D.3.

## 5.4 DEPTH ANALYSIS

To evaluate how REFORM affects internal representations across depth, we analyze four key metrics: SVCCA, mean attention entropy (MAE), effective rank, and Log-Det covariance. Together, these provide a multifaceted view of the model's representational dynamics.

**SVCCA.** Figure 4 shows REFORM reduces SVCCA similarity across middle layers, suggesting increased diversity and reduced redundancy. It increases similarity between the final and shallow layers, indicating early signals are preserved and propagated into later stages. Taken together, these trends suggest REFORM may help alleviate the bottleneck by maintaining intermediate diversity while preserving semantic continuity into the final layer.

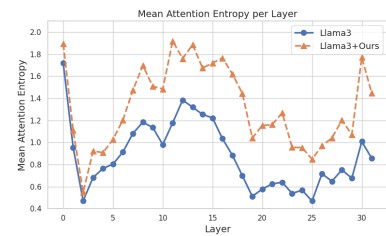

Figure 5: Layer-wise MAE in Llama3 and Llama3+Ours. Additional examples are provided in Appendix A.

**MAE.** Figure 5 compares layer-wise MAE values between the original Llama3 (blue) and REFORM (orange). REFORM tends to show higher MAE across layers, suggesting a broader attention spread and potentially diversity, particularly in mid-depth layers where semantic abstraction is most critical.

**Effective Rank & Log-Det covariance.** Figure 6 shows that REFORM maintains a slightly lower effective rank across middle layers compared to Llama3, indicating mid-depth compression and more concentrated variance. Toward the final layers, however, REFORM exhibits a notable increase in rank,

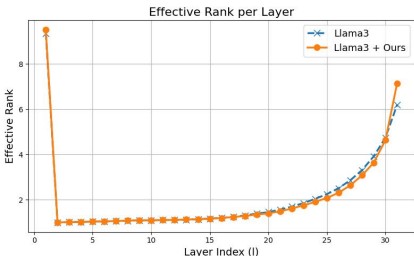
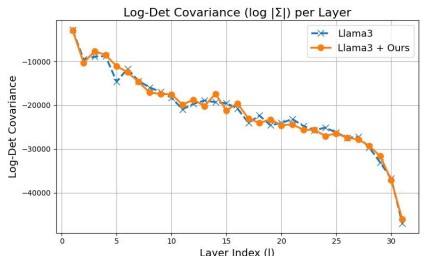

Figure 6: Layer-wise Effective Rank comparison between Llama3 and Llama3+Ours.

Figure 7: Log-Det covariance across Layers in Llama3 vs. Llama3+Ours.

ultimately surpassing Llama3 at the output layer. Since the final decoding stage critically depends on the expressiveness of the last-layer representation, this increase is particularly important, aligning with our design objective of enhancing final-layer diversity. As shown in Figure 7, the Log-Det covariance exhibits a similar trend, with REFORM showing a slight increase across middle layers and a consistent upward near the final layer. REFORM also shows stronger depth-wise improvements on Qwen2, Phi-3.5, and Mistral models, as detailed in Appendix D.4. For consistency and evaluation, we center our main analysis on Llama3, which serves as a challenging baseline.

**Overall Interpretation.** Taken together, REFORM mitigates the representational bottleneck of standard residual accumulation by promoting broader attention dispersion (MAE), recovering layer-wise diversity (SVCCA), building a compress–then–expand structure in the representation space (Effective Rank), and maintaining information spread across depth (Log-Det covariance matrix). In particular, REFORM encourages selective compression in middle layers while restoring expressiveness at the final layer. Rather than simply accumulating features, REFORM explicitly organizes inter-layer information flow, leading to structurally diverse and coherent final representations.

## 5.5 DISCUSSION

This work revisits the representation bottleneck in Transformers both theoretically and empirically. We show that residual connection tend to amplify magnitude without expanding semantic directions, resulting in stagnated diversity across depth. This aligns with the information bottleneck perspective, where the absence of selective compression leads to suboptimal representation learning.

Empirically, REFORM reduces middle layer redundancy, as reflected in lower SVCCA saturation and improved early-to-late alignment, corroborated by other probing metrics such as MAE and effective rank. On structurally complex tasks such as ARC-c, AQuA, and SVAMP, REFORM yields consistent gains across models. In contrast, we observe slight drops in specific model-task pairs on OBQA (Qwen2), WinoGrande (Llama3), and AddSub (Qwen2), suggesting potential limitations in scenarios with low-entropy outputs or minimal reasoning depth.

These patterns indicate that REFORM's uniform integration strategy, though generally effective, may not adapt optimally to all inference profiles. In particular, added abstraction may interfere with accurate predictions in tasks that mainly require simple or binary decisions. Future work could explore task-aware modulation mechanisms that dynamically adjust integration granularity, such as confidence-gated layer selection or depth-aware routing, based on task complexity and uncertainty.

## 6 CONCLUSION

We revisit the representation bottleneck in Transformers from an information bottleneck perspective, hypothesizing that residual connection accumulate redundant directions and limit representational diversity. This may restrict the model's ability to form compact intermediate abstractions. To probe this, we introduce REFORM—a training-time module that hierarchically integrates intermediate representations for more efficient encoding. REFORM consistently improves reasoning performance, illustrating how information-theoretic principles can inform compression-aware inter-layer design.

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

# A    MEAN ATTENTION ENTROPY RESULTS FROM VARIOUS PROMPT EXAMPLES

We visualize additional mean attention entropy (MAE) results obtained from various prompt examples. Figure 8 compares the MAE values of Llama1, Llama2, and Llama3, while Figure 9 shows the comparison between Llama3 and our proposed Llama3+Ours model.

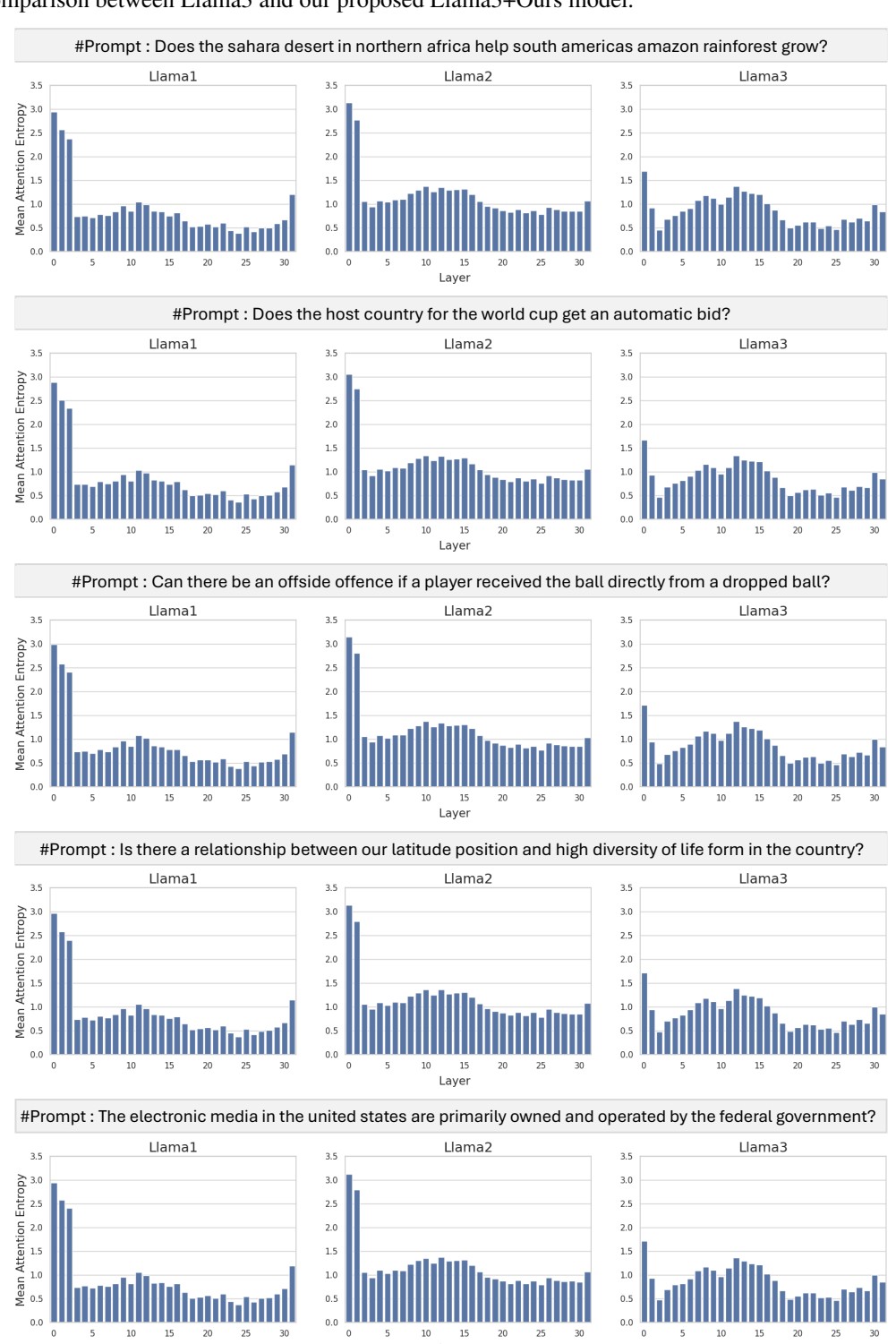

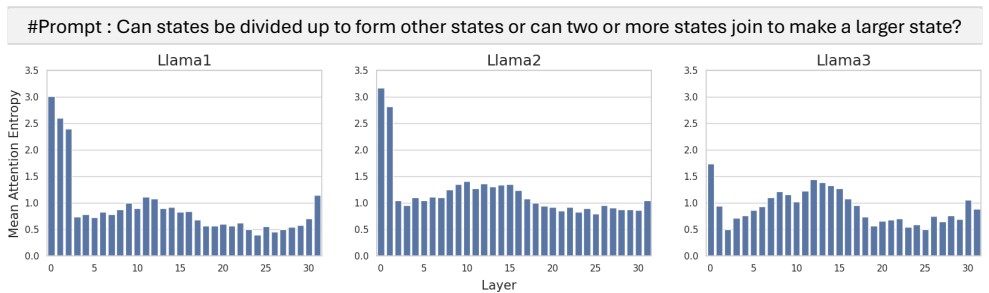

Figure 8: Mean attention entropy (MAE) across all layers for Llama1, Llama2, and Llama3.

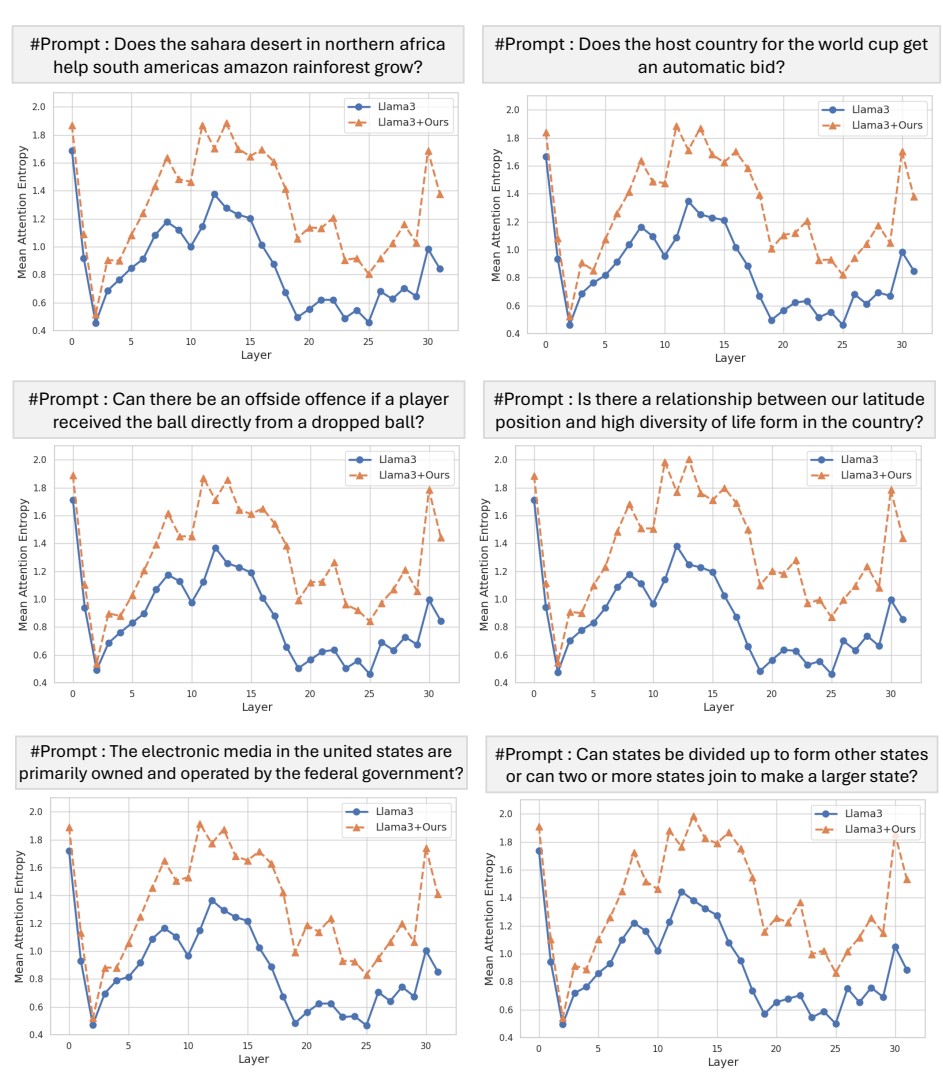

Figure 9: Mean attention entropy (MAE) comparison between Llama3 and Llama3+Ours.

# B PROOFS AND TECHNICAL DETAILS

**Support-restricted operators.** We distinguish two uses of the superscript "+":

(i) On *functions*, it indicates restriction to the support. For a PSD matrix $\Sigma$ with projector $P := \Sigma\Sigma^+$ onto $\mathrm{supp}(\Sigma)$,

$$\mathrm{Tr}^+(\Sigma) := \mathrm{Tr}(P\Sigma P) = \sum_{\lambda_i > 0} \lambda_i, \qquad \log\overset{+}{\det}(\Sigma) := \sum_{\lambda_i > 0} \log\lambda_i.$$

We also write $\lambda_{\min}^+(\Sigma)$ for the smallest positive eigenvalue.

(ii) On *matrices*, $X^+$ denotes the Moore–Penrose pseudoinverse.

Unless otherwise noted, all $\mathrm{Tr}^+$ and $\log\det^+$ below are taken on $\mathrm{supp}(\Sigma_{l-1})$ via $P_{l-1} := \Sigma_{l-1}\Sigma_{l-1}^+$.

**Conventions.** Notation and assumptions follow §3 (A1–A3, support stability). Let $P_{l-1} := \Sigma_{l-1}\Sigma_{l-1}^+$ be the orthogonal projector onto $\mathrm{supp}(\Sigma_{l-1})$, and set

$$r := \mathrm{rank}(\Sigma_{l-1}) = \dim\big(\mathrm{supp}(\Sigma_{l-1})\big).$$

All traces and determinants below are taken on $\mathrm{supp}(\Sigma_{l-1})$. For brevity write $H := H^{(l-1)}$.

**Local linearization with controlled remainder.** For each $l \in L$,

$$F^{(l)}(x) = A_l x + \varepsilon_l(x),$$

with $A_l = \rho_l I + E_l$, $\epsilon_l := \|A_l\|_2 \ll 1$, $\eta_l := \|E_l\|_2 \ll 1$. The nonlinear remainder satisfies

$$\left\| \mathbb{E}\big[\varepsilon_l(H)\,\varepsilon_l(H)^\top\big] \right\|_2 \leq c_\varepsilon\,\epsilon_l^2\,\|\Sigma_{l-1}\|_2.$$

**Alignment scalar.** Define

$$\rho_l := \frac{\mathbb{E}\langle H,\, F^{(l)}(H)\rangle}{\mathbb{E}\|H\|_2^2}.$$

Using $F^{(l)}(x) = A_l x + \varepsilon_l(x)$,

$$\rho_l = \frac{\mathrm{Tr}(A_l\Sigma_{l-1})}{\mathrm{Tr}\,\Sigma_{l-1}} + O(\epsilon_l^2),$$

where the $O(\epsilon_l^2)$ term follows from

$$\big|\mathbb{E}\langle H, \varepsilon_l(H)\rangle\big| \leq \big(\mathbb{E}\|H\|_2^2\big)^{1/2}\left(\mathrm{Tr}\,\mathbb{E}\big[\varepsilon_l(H)\varepsilon_l(H)^\top\big]\right)^{1/2} = O\big(\epsilon_l^2\,\mathbb{E}\|H\|_2^2\big).$$

### B.1 COVARIANCE RECURSION AND PERTURBATION SIZE

By definition $\Sigma_l = \mathbb{E}[H^{(l)}H^{(l)\top}]$ with

$$H^{(l)} = H + F^{(l)}(H).$$

Substituting $F^{(l)}(x) = A_l x + \varepsilon_l(x)$ gives

$$\Sigma_l = \mathbb{E}\big[(I + A_l)HH^\top(I + A_l)^\top\big] + \mathbb{E}\big[\varepsilon_l(H)\varepsilon_l(H)^\top\big]$$
$$+ \mathbb{E}\big[(I + A_l)H\,\varepsilon_l(H)^\top\big] + \mathbb{E}\big[\varepsilon_l(H)H^\top(I + A_l)^\top\big].$$

The first term expands as

$$(I + A_l)\Sigma_{l-1}(I + A_l)^\top = (1 + \rho_l)^2\Sigma_{l-1} + (1 + \rho_l)\big(E_l\Sigma_{l-1} + \Sigma_{l-1}E_l^\top\big) + E_l\Sigma_{l-1}E_l^\top.$$

Grouping the cross-terms and the nonlinear remainder into $\Delta_l$ yields

$$\Sigma_l = (1 + \rho_l)^2\Sigma_{l-1} + \Delta_l,$$

with, for a uniform constant $C > 0$,

$$\|\Delta_l\|_2 \leq C\big((1 + \rho_l)\eta_l + \eta_l^2 + \epsilon_l^2\big)\|\Sigma_{l-1}\|_2,$$

uniformly for $l \in L$. This isolates the main isotropic scaling via $\rho_l$ from the higher-order perturbations due to $E_l$ and the nonlinear remainder.

## B.2 SUPPORT-RESTRICTED FACTORIZATION AND THE PARAMETER $c$

All traces and $\log \det^+$ are taken on $\mathrm{supp}(\Sigma_{l-1})$. Define

$$M_l := (\Sigma_{l-1}^+)^{1/2} \Delta_l (\Sigma_{l-1}^+)^{1/2}, \quad \Sigma_l = (1 + \rho_l)^2 \Sigma_{l-1}^{1/2} (I + M_l) \Sigma_{l-1}^{1/2}.$$

Then

$$c := \|M_l\|_2 \ \leq \ C\left((1 + \rho_{\max})\eta_l + \eta_l^2 + \epsilon_l^2\right) \frac{\|\Sigma_{l-1}\|_2}{\lambda_{\min}^+(\Sigma_{l-1})}, \qquad c < 1. \tag{15}$$

## B.3 FIRST-ORDER CANCELLATION

**Lemma 1.** *If* $\mathrm{Tr}\left(P_{l-1} \mathrm{sym}(E_l)\right) = 0$, *where* $\mathrm{sym}(E_l) := \frac{1}{2}(E_l + E_l^\top)$, *then*

$$\mathrm{Tr}\ M_l = \mathrm{Tr}(E_l^\top P_{l-1} E_l) \ + \ O\left(\tfrac{r}{\tau}\epsilon_l^2\right) = O\left(\tfrac{r}{\tau}\eta_l^2\right) \ + \ O\left(\tfrac{r}{\tau}\epsilon_l^2\right).$$

## B.4 LOG-PSEUDO-DETERMINANT EXPANSION

If $c < 1$,

$$\log \overset{+}{\det}(\Sigma_l) = \log \overset{+}{\det}(\Sigma_{l-1}) + 2r \log(1 + \rho_l) + \mathrm{Tr} \log(I + M_l),$$

with the first-order term separated as

$$\mathrm{Tr} \log(I + M_l) = \mathrm{Tr}(M_l) + R_l, \qquad |R_l| \leq \frac{r\,c^2}{2(1 - c)}.$$

## B.5 EFFECTIVE-RANK STABILITY

Loewner sandwich $(1 - c)I \preceq I + M_l \preceq (1 + c)I$ and

$$\left| \mathrm{Tr}^+(\Sigma_l) - (1 + \rho_l)^2 \mathrm{Tr}^+(\Sigma_{l-1}) \right| = \left| \mathrm{Tr}(\Sigma_{l-1} M_l) \right| \leq c\ \mathrm{Tr}^+(\Sigma_{l-1})$$

imply, for normalized spectra $p, \tilde{p}$, that $\|\tilde{p} - p\|_1 \leq \frac{2c}{1-c}$ and $|\mathcal{H}(\tilde{p}) - \mathcal{H}(p)| \leq \frac{2c}{1-c} \log r$. Hence

$$r_{\mathrm{eff}}(\Sigma_l) = r_{\mathrm{eff}}(\Sigma_{l-1})\left(1 + O(c \log r)\right).$$

## B.6 MAIN PER-LAYER BOUNDS (EFFECTIVE SECOND ORDER)

Under §3 and $c < 1$,

$$\log \overset{+}{\det}(\Sigma_l) = \log \overset{+}{\det}(\Sigma_{l-1}) + 2r \log(1 + \rho_l) + O(rc^2), \qquad r_{\mathrm{eff}}(\Sigma_l) = r_{\mathrm{eff}}(\Sigma_{l-1})\left(1 + O(c \log r)\right),$$

with $c$ as in equation 15.

## B.7 ROBUSTNESS AND FAILURE MODES

Isotropic $A_l = \alpha I$ leaves $r_{\mathrm{eff}}$ invariant and shifts $\log \det^+$ by $2r \log(1 + \alpha)$. With $\mathrm{Tr}(P_{l-1} E_l) = 0$, shape changes are $O(\eta_l^2)$ and the residual in $\mathrm{Tr} \log$ is $O(rc^2)$. If $\lambda_{\min}^+ \downarrow 0$ (rank flip), $\rho_l \to -1$ (skip cancellation), or the remainder control fails, then $c \not< 1$ and $O(1)$ changes can occur.

**Constants.** All $O(\cdot)$ constants depend only on $(D, \tau, \delta)$, uniformly over $l \in L$.

# C DATASET AND HYPERPARAMETER DETAILS

## C.1 COMMONSENSE REASONING

We first trained the model using 170k training samples, and then evaluated the fine-tuned model on seven commonsense reasoning question-answering benchmarks. These benchmarks datasets are designed to assess models' inference skills through complex and nuanced question sets. Hyperparameter is provided in Table 5.

- **OBQA** (Mihaylov et al., 2018): a dataset composed of questions that demand multi-step reasoning, external commonsense knowledge, and deep reading comprehension. The test set comprises 500 items.

- **ARC-c/e** (Clark et al., 2018): includes both Challenge and Easy subsets from the ARC benchmark, featuring genuine grade-school-level multiple-choice science questions. The test sets contain 2376 and 1172 questions, respectively.

- **WinoG. (WinoGrande)** (Sakaguchi et al., 2021): a binary-choice task that assesses commonsense reasoning by requiring models to choose the correct completion for a given sentence. It includes 1267 test samples.

- **PIQA** (Bisk et al., 2019): consists of physically grounded commonsense questions, where each item offers two plausible solutions. There are 1830 samples in the test set.

- **BoolQ** (Clark et al., 2019): a binary question answering benchmark consisting of naturally occurring yes/no questions derived from unprompted queries. The test set contains 3,270 samples.

- **HellaS. (HellaSwag)** (Zellers et al., 2019): focuses on commonsense natural language inference, providing a context and several possible sentence completions. The test set includes 10,042 instances.

Table 5: Hyperparameter settings for LoRA and Ours on the Commonsense Reasoning.

| Hyperparameter | LoRA | Ours |
|---|---|---|
| $\lambda_{orth}$ | – | 0.0001 - 0.05 - 0.0001 |
| $\lambda_{ca}$ | – | 0.1 |
| $\lambda_{cs}$ | – | 0.0001 - 0.05 - 0.01 |
| Rank $r$ | | 16 |
| $\alpha$ | | 32 |
| Dropout | | 0.05 |
| Batch size | | 32 |
| Micro batch size | | 16 |
| Epochs | | 3 |
| Learning rate | | 3e-4 |
| Target module | | q, k, v, up, down |

## C.2 ARITHMETIC REASONING

For arithmetic reasoning evaluation, we consider five benchmarks. These datasets span a wide range of mathematical problems, from basic arithmetic operations to complex multi-step reasoning tasks. The model was fine-tuned on 10K training samples to comprehensive assess its numerical reasoning capabilities. Hyperparameter is provided in Table 6.

- **AddSub** (Hosseini et al., 2014): a dataset composed of addition and subtraction-based word problems for elementary-level learners. It includes 395 test samples.

- **SVAMP** (Patel et al., 2021): single-variable arithmetic word problems adapted from an existing dataset by applying minor textual modifications, aimed at up-to-4th grade students. It includes 1,000 problems in the test set.

- **AQuA** (Ling et al., 2017): a benchmark of algebraic word problems accompanied by natural language rationales, designed to evaluate both answer accuracy and reasoning explainability. The test set includes 254 questions.

- **MultiArith** (Roy & Roth, 2015): math word problems that require multiple arithmetic operations and reasoning steps to solve. There are 600 problems in the test set.

- **GSM8K** (Cobbe et al., 2021): a collection of high-quality, linguistically diverse math word problems designed by human authors, targeting grade school level arithmetic reasoning. The test set contains 1,319 questions.

Table 6: Hyperparameter settings for LoRA and ours on the Arithmetic Reasoning.

| Hyperparameter | LoRA | Ours |
|---|---|---|
| $\lambda_{orth}$ | – | 0.0001 - 0.05 - 0.0001 |
| $\lambda_{ca}$ | – | 0.1 |
| $\lambda_{cs}$ | – | 0.0001 - 0.05 - 0.01 |
| Rank $r$ | | 16 |
| $\alpha$ | | 32 |
| Dropout | | 0.05 |
| Batch size | | 32 |
| Micro batch size | | 16 |
| Epochs | | 3 |
| Learning rate | | 3e-4 |
| Cutoff length | | 256 |
| Validation set size | | 120 |
| Target module | | q, k, v, up, down |

## D ADDITIONAL EMPIRICAL RESULTS

### D.1 BENCHMARK RESULTS WITH STANDARD DEVIATIONS

Tables 7 and 8 present the empirical results on commonsense reasoning and arithmetic reasoning benchmarks. Each score indicates the mean and standard deviation computed over five random seeds.

Table 7: Empirical results on Commonsense Reasoning. Each score denotes the mean and standard deviation (in parentheses) across five random seeds.

| Method | OBQA | ARC-e | ARC-c | WinoG. | PIQA | BoolQ | HellaS. | Avg. |
|---|---|---|---|---|---|---|---|---|
| Llama3 + LoRA | 84.9 ($\pm$ 0.6) | 90.2 ($\pm$ 0.2) | 79.9 ($\pm$ 0.6) | **85.5** ($\pm$ 0.3) | 87.9 ($\pm$ 0.6) | 69.9 ($\pm$ 0.9) | 94.4 ($\pm$ 0.2) | 84.7 ($\pm$ 0.2) |
| + Ours | **85.3** ($\pm$ 0.4) | **90.3** ($\pm$ 0.1) | **80.5** ($\pm$ 0.4) | 85.1 ($\pm$ 0.7) | **88.7** ($\pm$ 0.5) | **71.8** ($\pm$ 0.8) | **94.9** ($\pm$ 0.1) | **85.2** ($\pm$ 0.2) |
| Qwen2 + LoRA | **89.3** ($\pm$ 0.6) | 92.7 ($\pm$ 0.7) | 83.1 ($\pm$ 0.4) | 85.5 ($\pm$ 0.4) | 89.5 ($\pm$ 0.3) | 73.9 ($\pm$ 0.6) | **94.6** ($\pm$ 0.1) | 86.9 ($\pm$ 0.3) |
| + Ours | 89.1 ($\pm$ 0.4) | **93.2** ($\pm$ 0.2) | **84.2** ($\pm$ 0.4) | **85.7** ($\pm$ 0.6) | **89.8** ($\pm$ 0.4) | **74.1** ($\pm$ 0.3) | 94.8 ($\pm$ 0.2) | **87.3** ($\pm$ 0.2) |
| Mistral + LoRA | 84.0 ($\pm$ 0.6) | 85.7 ($\pm$ 0.6) | 73.4 ($\pm$ 0.6) | 83.0 ($\pm$ 0.6) | 87.4 ($\pm$ 0.7) | 71.0 ($\pm$ 1.0) | 90.3 ($\pm$ 0.3) | 82.1 ($\pm$ 0.5) |
| + Ours | **85.4** ($\pm$ 0.7) | **86.1** ($\pm$ 0.4) | **74.6** ($\pm$ 0.6) | **84.5** ($\pm$ 0.4) | **87.5** ($\pm$ 0.6) | **71.8** ($\pm$ 0.4) | **90.7** ($\pm$ 0.9) | **82.9** ($\pm$ 0.6) |
| Phi-3.5 + LoRA | 88.6 ($\pm$ 0.7) | 94.4 ($\pm$ 0.3) | 85.4 ($\pm$ 0.3) | 83.0 ($\pm$ 0.6) | 86.1 ($\pm$ 0.6) | 70.3 ($\pm$ 0.4) | **90.8** ($\pm$ 0.3) | 85.5 ($\pm$ 0.3) |
| + Ours | **89.3** ($\pm$ 0.5) | **94.7** ($\pm$ 0.3) | **86.0** ($\pm$ 0.3) | **83.6** ($\pm$ 0.3) | **86.8** ($\pm$ 0.3) | **70.5** ($\pm$ 0.4) | **90.8** ($\pm$ 0.4) | **86.0** ($\pm$ 0.4) |

Table 8: Empirical results on Arithmetic reasoning. Each score denotes the mean and standard deviation (in parentheses) across five random seeds.

| Method | SVAMP | AddSub | AQuA | MultiArith | GSM8K | Avg. |
|---|---|---|---|---|---|---|
| Llama3 + LoRA | 82.7 ($\pm$ 0.5) | **92.9** ($\pm$ 1.5) | 32.3 ($\pm$ 0.9) | 98.0 ($\pm$ 0.3) | 72.9 ($\pm$ 0.7) | 75.8 ($\pm$ 0.6) |
| + Ours | **83.8** ($\pm$ 0.3) | **92.9** ($\pm$ 1.0) | **34.3** ($\pm$ 0.5) | **98.3** ($\pm$ 0.3) | **73.6** ($\pm$ 0.5) | **76.6** ($\pm$ 0.4) |
| Qwen2 + LoRA | 82.8 ($\pm$ 0.7) | **91.6** ($\pm$ 0.4) | 36.9 ($\pm$ 1.1) | **97.3** ($\pm$ 0.7) | 73.9 ($\pm$ 0.5) | 76.5 ($\pm$ 0.7) |
| + Ours | **83.5** ($\pm$ 0.4) | 91.2 ($\pm$ 0.4) | **37.6** ($\pm$ 0.4) | **97.3** ($\pm$ 0.3) | **75.0** ($\pm$ 0.6) | **76.9** ($\pm$ 0.6) |
| Mistral + LoRA | 65.3 ($\pm$ 1.5) | 87.9 ($\pm$ 0.7) | 23.7 ($\pm$ 2.0) | 95.0 ($\pm$ 0.6) | 62.6 ($\pm$ 0.6) | 66.9 ($\pm$ 1.5) |
| + Ours | **66.8** ($\pm$ 1.2) | **88.9** ($\pm$ 0.7) | **24.7** ($\pm$ 1.6) | **95.3** ($\pm$ 0.4) | **63.2** ($\pm$ 0.4) | **67.8** ($\pm$ 1.0) |
| Phi-3.5 + LoRA | 76.4 ($\pm$ 0.3) | **86.1** ($\pm$ 0.7) | 33.6 ($\pm$ 0.7) | 98.1 ($\pm$ 0.4) | 76.0 ($\pm$ 0.2) | 74.0 ($\pm$ 0.3) |
| + Ours | **76.9** ($\pm$ 0.3) | 85.5 ($\pm$ 0.8) | **34.1** ($\pm$ 0.4) | **98.4** ($\pm$ 0.4) | **77.0** ($\pm$ 0.4) | **74.4** ($\pm$ 0.3) |

### D.2 ADDITIONAL ABLATION STUDY

**Effect of the number of local and global groups.** We conduct an ablation study to examine how the granularity of local and global grouping within the aggregator influences model performance. The experiment is conducted on the Llama3 architecture using the arithmetic reasoning benchmark. As shown in Table 9, the best performance (76.6%) is achieved when the number of local groups is set to 4 and global groups to 8. This configuration effectively balances fine-grained inter-layer modeling through local grouping and broad semantic abstraction via global integration. In contrast, overly ag-

Table 9: Ablation study on the number of local and global groups with different overlapping windows.

| #Local Groups | #Global Groups | Overlapping Window | Average Performance (%) |
|---|---|---|---|
| 2 | 16 | – | 76.5 |
| 4 | 8 | – | **76.6** |
| 8 | 4 | – | 76.2 |
| 16 | 2 | – | 75.7 |
| 4 | 14 | 1 | 76.1 |
| 8 | 6 | 4 | 76.3 |
| 16 | 3 | 8 | 75.2 |

gressive local aggregation (e.g., 16 local groups) results in degraded performance (75.7%), suggesting that excessively compressed groups may obscure meaningful layer-wise transitions, thereby hindering hierarchical abstraction. We also experimented with a sliding-window (overlapping) grouping scheme, but its performance was consistently lower (e.g., 76.1%–76.3%), indicating that overlapping adds complexity without improving abstraction quality. These results highlight the importance of maintaining an appropriate staging resolution when integrating multi-depth representations.

Table 10: Ablation on auxiliary objectives.

| Objective | Avg. |
|---|---|
| Llama3 + LoRA | 75.7 |
| w/ $\mathcal{L}_{orth}$ | 75.6 |
| w/ $\mathcal{L}_{ca}$ | 75.7 |
| w/ $\mathcal{L}_{cs}$ | **76.0** |

**Effect of auxiliary objectives.** Table 10 reports the contribution of individual objectives. Among them, $\mathcal{L}_{cs}$ provides the most notable gain, since it helps preserve global representation quality even when the auxiliary module is disabled at inference. In contrast, $\mathcal{L}_{orth}$ and $\mathcal{L}_{ca}$ show limited or negligible improvements, sometimes even slightly reducing performance, suggesting that their effects are secondary compared to the consistency regularization offered by $\mathcal{L}_{cs}$.

### D.3 QUALITATIVE RESULTS

We conduct qualitative analyses across various models and benchmarks to gain deeper insight into the effects of REFORM on representation behavior. In the following examples, correct predictions are highlighted in green, while incorrect ones are marked in red. Each case is selected to illustrate specific strengths and limitations of our method in handling semantic abstraction, multi-step reasoning, and syntactic cues.

Table 11 illustrates a limitation of semantic abstraction. While REFORM tends to rely on high-level semantic cues (e.g.,"Sarah is better"), the task instead requires reasoning based on functional roles within a comparative structure. The incorrect prediction indicates that REFORM's abstraction mechanism, although effective for capturing semantic integration, may overlook shallow syntactic cues, such as role alignment and comparative reversal. These cues are essential for resolving referential ambiguity in structurally constrained contexts.

Table 12 highlights REFORM's strength in multi-step reasoning tasks that require structured abstraction. In tasks involving proportional energy distribution, temporal causality, or hypothetical scenarios, REFORM consistently outperforms the original model by preserving latent reasoning chains across steps. While the original model selects a semantically plausible but incorrect answer, REFORM accurately identifies the correct answer by maintaining intermediate reasoning related to kinetic energy transformation. These results support REFORM's design objective: to preserve structured semantic trajectories through selective compression, particularly in tasks that requiring multi-stage causal or proportional alignment.

Table 13 illustrates a representative failure case that highlights a trade-off in REFORM's abstraction strategy. In this example, the question asks for the total number of turnips, but REFORM mistakenly

includes pumpkins in the final answer. This suggests that although REFORM supports conceptual integration, it can blur task-specific word boundaries when the question requires strict attention to surface-level details. The example emphasizes a key limitation of abstraction-based models: while helpful for reasoning tasks that need high-level understanding, abstraction must be carefully controlled when accurate lexical interpretation is important.

Table 14 presents a case where REFORM demonstrates its strength in handling multi-step reasoning. The task requires understanding a hidden quantity transfer over time and correctly updating another person's state based on it. While the original model incorrectly sums the total number of cards and fails to isolate the relevant change, REFORM correctly infers the number of cards transferred and applies that change to Jeff's count. This example highlights how REFORM helps maintain structured reasoning over sequential steps and improves performance on tasks that involve causal or temporal dependencies.

### D.4    DEPTH-WISE REPRESENTATION IMPROVEMENTS ACROSS MODELS

Figures 10 to 15 present the effective rank and log-determinant of covariance across depth for Qwen2, Phi-3.5, and Mistral models. In all cases, REFORM consistently demonstrates improved depth-wise representation structure compared to the base models. Specifically, REFORM tends to suppress redundancy in the middle layers while enhancing representational richness near the output. This pattern aligns with our findings on Llama3 and further confirms the generalizability of REFORM's design across different model architectures.

Table 11: Qualitative results in WinoGrande benchmark.

| Base model : Llama3-8B-Instruct + LoRA | |
| --- | --- |
| Question | Sarah was a much better surgeon than Maria so _ always got the easier cases. |
| Options | Option1 : Sarah, Option2 : Maria |
| Original | the correct answer is option2. |
| REFORM | the correct answer is option1. |

Table 12: Qualitative results in ARC-C benchmark.

| Base model : Phi-3.5-Mini-Instruct + LoRA | |
| --- | --- |
| Question | An astronaut drops a 1.0 kg object and a 5.0 kg object on the Moon. Both objects fall a total distance of 2.0 m vertically. Which of the following best describes the objects after they have fallen a distance of 1.0 m? |
| Answer | Answer1: They have each lost kinetic energy. Answer2: They have each gained the same amount of potential energy. Answer3: They have each lost the same amount of potential energy. Answer4: They have each gained one-half of their maximum kinetic energy. |
| Original | the correct answer is answer3. |
| REFORM | the correct answer is answer4. |

Table 13: Qualitative results in AddSub benchmark.

| Base model : Qwen2-7B-Instruct + LoRA | |
| --- | --- |
| Question | Sally grew 113 turnips and 118 pumpkins . Mary grew 129 turnips . How many turnips did they grow in total? |
| Original | To find the total number of turnips Sally and Mary grew, we need to add the number of turnips each of them grew. Step 1: Add the number of turnips Sally grew to the number of turnips Mary grew. 113 + 129 = 242. Step 2: The total number of turnips they grew is 242. Therefore, Sally and Mary grew a total of 242 turnips. Prediction : 242.0 |
| REFORM | Step 1: Add the number of turnips Sally grew to the number of turnips Mary grew. 113 + 129 = 242. Step 2: Add the number of pumpkins Sally grew to the number of pumpkins Mary grew. 118 + 0 = 118. Step 3: Add the total number of turnips to the total number of pumpkins. 242 + 118 = 360. Therefore, they grew a total of 360 turnips. The answer in Arabic numerals is 360. Prediction : 360.0 |

Table 14: Qualitative results in SVAMP benchmark.

| Base model : Mistral-7B-Instruct-v0.3 + LoRA | |
| --- | --- |
| Question | Nell collects baseball cards. She had 566 cards while Jeff had 234 cards. She gave some of her cards to Jeff and now has 535 cards left. How many cards does Jeff have now? |
| Original | Step 1: Find the total number of cards Nell and Jeff had before Nell gave some to Jeff. 566 + 234 = 800. Step 2: Find the number of cards Nell gave to Jeff. 800 - 535 = 265. Step 3: Add the number of cards Nell gave to Jeff to the number of cards Jeff had before. 234 + 265 = 499. Therefore, Jeff now has 499 cards. The answer in Arabic numerals is 499. Prediction : 499.0 |
| REFORM | Nell gave away 566 - 535 = 31 cards to Jeff. So, Jeff now has 234 + 31 = 265 cards. Therefore, Jeff has 265 cards now. The answer in Arabic numerals is 265. Prediction : 265.0 |

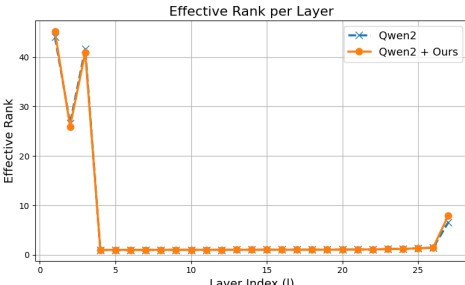

Figure 10: Layer-wise Effective Rank comparison between Qwen2 and Qwen2+Ours.

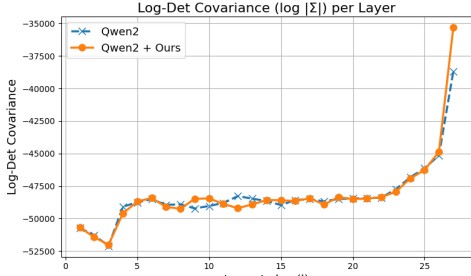

Figure 11: Log-Det covariance across Layers in Qwen2 vs. Qwen2+Ours.

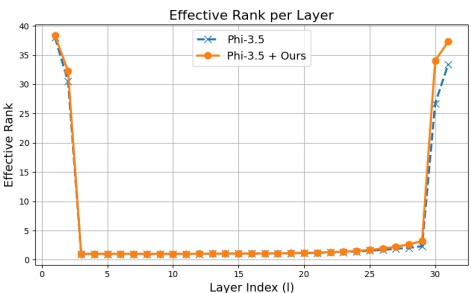

Figure 12: Layer-wise Effective Rank comparison between Phi-3.5 and Phi-3.5+Ours.

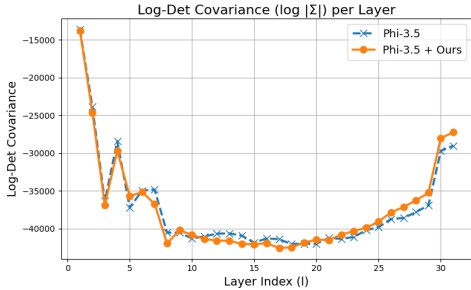

Figure 13: Log-Det covariance across Layers in Phi-3.5 vs. Phi-3.5+Ours.

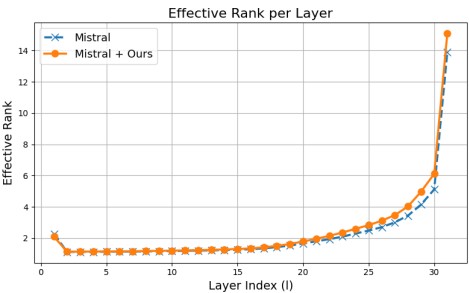

Figure 14: Layer-wise Effective Rank comparison between Mistral and Mistral+Ours.

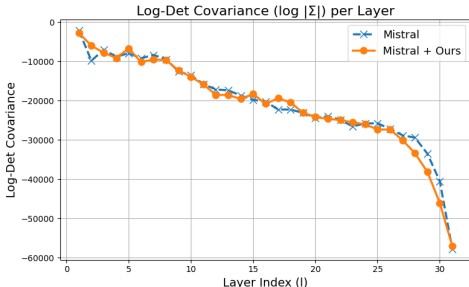

Figure 15: Log-Det covariance across Layers in Mistral vs. Mistral+Ours.