# OpenReview forum: "REFORM : Residual Filtering through Neural Aggregators for Layer-Wise Representation Integrity"
_ICLR.cc/2026/Conference — ICLR 2026 Conference Withdrawn Submission_

### Official Review · Reviewer_du3h · 2025-10-24

**Soundness:** 3
**Presentation:** 3
**Contribution:** 3
**Rating:** 4
**Confidence:** 3

**Summary:**

This paper focuses on the structural limitations of residual accumulation on deep representations in Transformers, pointing out that indiscriminately preserving signals leads to a “representation bottleneck” in the middle-to-late layers; the authors provide an information-theoretic perspective and propose a lightweight module, REFORM, that is used only during training and removed at inference: “local aggregation” preserves inter-layer continuity and “global fusion” achieves semantic abstraction, supplemented by three auxiliary objectives—correlation alignment, orthogonality constraint, and cosine similarity—to restructure intermediate representations; the module is dismantled during inference, thereby incurring no runtime overhead. Multiple open-source LLMs are reportedly improved consistently but mildly on common-sense and mathematical reasoning benchmarks, while analyses based on SVCCA, attention entropy (MAE), and effective rank indicate that REFORM can cultivate richer representations in the middle-to-late layers, thereby alleviating the structural limitation of the residual path with minimal inter-layer aggregation.

**Strengths:**

1. The problem is clearly defined and important: it systematically points out that “residual accumulation” forms a representation bottleneck in deep layers and explains the mechanism of “amplifying magnitude without expanding semantic directions” from an information-theoretic/spectral perspective; the motivation and phenomena are well described.

2. The structural design is simple and intuitive: it proposes hierarchical aggregation of “local first, then global”: within-group fine-grained selection (Local), and cross-group semantic abstraction with 1D Conv (Global); the two-stage intuition of “select first, then abstract” is clear.

3. The analysis metrics are diverse and consistent with the motivation: SVCCA, attention entropy (MAE), effective rank, and Log-Det reveal the representation dynamics of “mid-layer compression—final-layer re-expansion,” mutually corroborating the theoretical narrative.

**Weaknesses:**

1. Overall numerical gains are relatively mild: the method has a novel angle, but the gains are limited on several benchmarks. Evidence for a “universally significant improvement” remains insufficient.

2. Is this method more suitable for application in pre-training rather than in the fine-tuning stage?

3. Can the authors present the model performance when retaining the REFORM module during inference?

**Questions:**

N/A

---

### Official Review · Reviewer_P7F9 · 2025-10-28

**Soundness:** 2
**Presentation:** 2
**Contribution:** 2
**Rating:** 2
**Confidence:** 2

**Summary:**

The paper introduces REFORM, a training-time module that aims to mitigate representation bottlenecks in Transformers by promoting more diverse and disentangled layer-wise representations. The method aggregates hidden states across layers through a two-level hierarchy: local aggregation over adjacent layers and global aggregation over the full layer stack, producing auxiliary representations that complement the model’s final hidden states.

During training, REFORM is guided by three auxiliary objectives encouraging alignment, orthogonality, and consistency between the aggregated and original representations. These losses are designed to make the model learn richer intermediate features while ensuring stability once the module is removed at inference.

REFORM is evaluated across several LLM architectures on commonsense and arithmetic reasoning benchmarks, showing modest but consistent accuracy improvements and signs of increased representational diversity based on SVCCA and entropy metrics. The module incurs no inference-time cost, which supports its practical appeal.

**Strengths:**

1. Interesting conceptual motivation.
The idea of explicitly regularizing intermediate representations to mitigate redundancy between layers is meaningful and timely, connecting to active discussions on representational collapse, information bottlenecks, and interpretability in Transformers.

2. Theoretical framing of the residual bottleneck.
The paper makes a valuable attempt to formally characterize the residual bottleneck through mathematical and information-theoretic analysis, providing a principled foundation for the proposed architectural modifications.

3. Training-only auxiliary design.
Removing REFORM at inference time is appealing from a practical standpoint: it avoids additional inference cost while potentially improving training dynamics and generalization.

4. Variety of representational metrics.
The inclusion of different quantitative measures (MAE, SVCCA, effective rank, log-det covariance) shows an attempt to explore multiple facets of representational diversity rather than relying on a single proxy.

5. Clear ablation intent.
The paper does perform ablations across layer groups and some loss components, suggesting awareness of architectural sensitivity, even though the results are not fully consistent or clearly presented.

6. Potential to inspire follow-up work.
The overall framing - explicitly penalizing representational redundancy between Transformer layers - could inspire further research in regularization, model interpretability, or modular architectures.

**Weaknesses:**

1. Lack of clarity and consistency in metric introduction.
The paper begins by defining mean attention entropy (MAE) but immediately introduces other measures such as SVCCA and covariance-based metrics without clear definitions or context. Later, effective rank and log-determinant covariance appear with little explanation. As a result, it remains unclear how these metrics jointly quantify the “representation bottleneck” or why they are meaningful proxies for representational diversity.

2. Neglect of the MLP component.
The analysis and motivation focus almost entirely on the attention mechanism, with no discussion of the MLP sublayer in Transformers. It is unclear whether REFORM interacts with or aggregates representations that include MLP outputs, and if not, why this major component is omitted. The precise integration point of REFORM (after attention, after MLP, or after each full Transformer block) is never explicitly stated.

3. Overly technical and poorly structured mathematical exposition.
Section 3 presents dense, notation-heavy derivations without properly introducing symbols or providing intuition. Many variables are used before definition, and the section lacks high-level explanations linking the math to the problem of representational bottlenecks. This level of detail would be better suited for an appendix, as the current presentation interrupts readability.

4. Unclear and under-justified design choices.
The motivation for dividing 32 layers into eight groups of four is not justified. Why exactly four layers per group, and why non-overlapping groups instead of a sliding or adaptive window? Similarly, fixed coefficients such as λ_ca = 0.1 are stated without reasoning or ablation. Several design and hyperparameter decisions appear arbitrary rather than empirically motivated.

5. Ambiguous training-inference decoupling.
REFORM is removed entirely during inference, yet the auxiliary losses (especially the orthogonality objective) are said to encourage complementary features between REFORM outputs and base model representations. It remains unclear how this complementary information is retained once REFORM is detached. The conceptual consistency of a “training-only” module that purportedly improves representation quality at inference is not well explained.

6. Weak experimental rigor and limited interpretability of results.
Reported gains are small (often under 1%) and lack evidence of statistical significance. The number of test runs and sample sizes are not specified in the main text. Variance is only provided in the appendix, though it is crucial given the small margins. It is also unclear whether the “+Ours” models were trained from scratch or fine-tuned, which affects interpretability of the improvements. Figures 6-7 further illustrate this issue: differences between outputs are barely visible, yet the paper draws interpretive conclusions from them without statistical support.

7. Presentation and formatting issues.
Tables and figures are inconsistently placed and labeled. Tables 1 and 2 appear in the Experimental Setting section rather than Results. The caption for Table 4 is formatted too close to Table 3 above it, which makes it appear as if it belongs to Table 3, this spacing issue creates confusion. Figure captions lack sufficient explanation (especially Figure 3, which should include a clearer architectural overview, and could benefit from adding a legend). Additionally, references are inconsistently formatted, with arXiv citations following different styles. Overall, the visual and textual layout could be significantly improved for clarity.

8. Repetitive and unsynthesized discussion and conclusion.
The Discussion section primarily restates experimental findings without offering deeper analysis or synthesis. It could be merged with the Conclusion to improve cohesion and avoid redundancy.

**Questions:**

1. Could the authors clarify how REFORM interacts with MLP sublayers? Is the representational diversity objective applied only to attention outputs, or to full block outputs?

2. Since REFORM is removed at inference, what evidence supports that its training-induced representations retain complementary information rather than simply altering optimization dynamics?

3. What motivated the decision to group every 4 layers together? Did the authors test overlapping groups or varying group sizes, and how sensitive are results to this choice?

4. Reported gains are small (<1%). Could the authors provide variance over multiple runs or statistical significance tests to confirm reliability?

5. If the main motivation is representation disentanglement or diversity, could the authors visualize how internal feature spaces differ with and without REFORM?

---

### Official Review · Reviewer_FK4n · 2025-10-31

**Soundness:** 3
**Presentation:** 3
**Contribution:** 3
**Rating:** 6
**Confidence:** 3

**Summary:**

REFORM introduces a training-time-only residual filtering mechanism designed to alleviate representation bottlenecks in Transformer models caused by unselective residual accumulation. It employs two neural aggregators (local and global) that hierarchically integrate inter-layer representations. Three auxiliary losses — correlation alignment, orthogonality, and cosine similarity — regularize layer-wise consistency and ensure inference-time stability. The REFORM module is stochastically activated during training and removed entirely during inference, adding zero runtime cost. Empirical results on multiple LLMs show consistent gains across reasoning and arithmetic benchmarks. Analytical results demonstrate better representation diversity and structured information flow.

**Strengths:**

This work tackles the under-explored problem of residual bottlenecking in large-scale Transformers from an information-theoretic perspective, proposing a module that can be seamlessly integrated into any Transformer-based architecture without re-training from scratch, affects only training while leaving inference cost and architecture unchanged, and is supported by both theoretical and empirical analyses demonstrating improved representation diversity.

**Weaknesses:**

1. Although mitigated by L_cs, removing the REFORM module at inference time may still introduce subtle training–inference distribution shifts, especially as the context length increases. Such discrepancies could accumulate over longer sequences. Since the experiments in this paper mainly focus on commonsense reasoning and arithmetic reasoning tasks with relatively short contexts, it would be valuable to validate REFORM on longer and more complex contexts to assess its robustness.

2. The paper lacks comparisons with other recent approaches that also target improvements in residual connection design, such as DeepCrossAttention [1]. Including these baselines would strengthen the empirical claims and clarify REFORM’s relative contribution.

3. It remains unclear whether the additional REFORM module significantly increases training cost. The authors are encouraged to report GPU memory consumption and training speed compared to the baseline (e.g., standard LoRA fine-tuning). This would help readers evaluate the practical trade-offs of adopting REFORM.

[1] DeepCrossAttention: Supercharging Transformer Residual Connections, 2025.

**Questions:**

See Weaknesses

---

### Official Review · Reviewer_LX52 · 2025-11-04

**Soundness:** 2
**Presentation:** 2
**Contribution:** 2
**Rating:** 4
**Confidence:** 3

**Summary:**

This paper studies how cumulative residual connections in Transformer LLMs can indiscriminately pass signals forward, creating representation bottlenecks in deeper layers. The authors present an information-theoretic analysis of this issue and propose REFORM, a lightweight, training-only module that hierarchically aggregates hidden states across layers

**Strengths:**

1. The paper is well-written and easy to follow.
2. The paper provides both theoretical and empirical analyses, and the research problem is interesting.

**Weaknesses:**

1. My main concern is that the experimental improvements are not significant, which seems insufficient to support the paper’s motivation and conclusions.
2. I would be more inclined to accept the paper if it focused more on analyzing the representation bottleneck rather than proposing a new training method. REFORM adds considerable training overhead yet delivers only modest performance gains. Moreover, the analysis of the representation bottleneck feels insufficiently deep. After reading, I am still left with the following confusions:
    a. Is the so-called representation bottleneck issue actually harmful? I haven’t seen compelling evidence to support that.
    b. If I’m not mistaken, the authors’ argument in Figure 1 is that Llama3 exhibits greater representational capacity than Llama2 and Llama1, and that compared with Llama2 it displays more pronounced layer-wise variation in attention patterns; therefore, representational capacity relates not only to global entropy but also to layer-wise variation. This strikes me as weakly substantiated, as it shows correlation rather than causation.
3. I recommend including a comparison of training efficiency and GPU memory consumption.

**Questions:**

See Weaknesses

---

### Note · Authors · 2025-11-14

I have read and agree with the venue's withdrawal policy on behalf of myself and my co-authors.